# Effect of Eight Weeks of in Season Training with Wearable Resistance Attached to the Forearm on Spike Velocity in Female Volleyball Players

**DOI:** 10.3390/jfmk10040458

**Published:** 2025-11-21

**Authors:** Milosz Mielniczek, Patrick Lunde, Wojciech Grzyb, Roland van den Tillaar

**Affiliations:** 1Department of Sports Sciences and Physical Education, Nord University, 7600 Levanger, Norway; milosz.msg@gmail.com (M.M.);; 2Department of Physical Education, Gdansk University of Physical Education and Sports, 80-336 Gdansk, Poland; wojciech.grzyb@awf.gda.pl

**Keywords:** forearm loading, strength training, velocity, calf loading

## Abstract

**Objective:** The aim of the study was to investigate the effect of eight weeks of in-season training with wearable resistance attached to the forearm on spike velocity in female volleyball players. **Methods:** A total of 13 senior female volleyball players belonging to the same team (age 23.5 ± 3.2 years, body mass 66.8 ± 6.9 kg, and height 174.7 ± 5.8 cm) participated and were divided into an experimental group (*n* = 7) or a control group (*n* = 6). Both groups participated in the same training sessions, but the experimental group trained with wearable resistance attached to the forearm, while the control group wore wearable resistance on the calf during the training sessions. Before and after an eight-week training period, spike velocity was tested. **Results:** The main findings were that spike velocity did not increase in either group, which was contrary to our hypothesis. In fact, neither the forearm-loaded experimental group nor the calf-loaded control group showed any performance improvement; instead, both groups demonstrated small, non-significant declines in spike velocity over the 8-week period (on average, about 2.1% in both groups). These outcomes occurred with high individual variability, with no significant difference between the groups (time × group: *F*(1, 11) = 0.008, *p* = 0.929). The between-group contrast was trivial, reinforcing that the forearm-loading intervention offered no clear benefit over normal training in practical terms. **Conclusions:** Taken at face value, this protocol does not support using wearable resistance during the competitive season to enhance spike velocity. It may be that off-season or individualized-load protocols could elicit more positive effects when overall training and fatigue levels are better controlled.

## 1. Introduction

Spike velocity is a central performance factor in volleyball, influencing the likelihood of getting through the block or the defense [1]. Faster attacks increase attack efficacy by reducing defensive organization [2], and higher serve velocities significantly reduce the reception quality of the opposite team [3]. There are several strength training methods varying from general strength training to specific training to enhance spike velocity [4]. One of these specific strength training methods is the use of wearable resistance attached to the limbs [5,6,7].

Wearable resistance refers to light external loads attached to the body (e.g., sleeves or vests), allowing athletes to overload sport-specific movements during training [8]. Wearable resistance has been shown to positively influence key actions such as sprinting and change in direction, with systematic review evidence indicating clear benefits for sprint acceleration when loads are ≤10% of body mass [9,10]. However, much of the wearable resistance literature in team sports has focused on weighted vests and lower-body attachments, typically in sprinting or warm-up contexts [11,12,13]. In contrast, the application of using wearable resistance attached to the upper limb to improve performance is sparse. Three studies in overhead throwing sports have shown that the use of wearable resistance mounted to the upper arm had a positive effect on throwing performance [14,15,16]. The effects varied from 3.9 to 12% after 6 to 10 weeks of training in experienced and novice handball players [14,15,16]. As handball throwing has many biomechanically similarities with the volleyball spike [17], it is expected that similar positive results with wearable resistance training attached to the forearm would occur in the volleyball spike.

Nevertheless, wearable resistance in volleyball, particularly attached to the upper limb, remains underexplored. The present study addresses this gap by testing an in-season wearable resistance intervention in senior female volleyball players. The rationale is rooted in the specificity principle: adaptations are greatest when training closely resembles the target skill in force–time and coordination demands [18]. Attaching wearable resistance to a limb increases its resistance to movement, raising joint torque demands and potentially altering coordination [19]. Distal loading, such as the forearm, magnifies rotational inertia and may alter swing velocity and end-effector velocity, while small loads may provide useful overload and heavier loads risk disruption [8,20]. Taken together, these mechanisms suggest that wearable resistance may involve both benefits and trade-offs, rather than guaranteeing improvements in spike velocity.

During the competitive season, players are exposed to high and frequent loads from matches, travel, and limited recovery time, which can slow down their performance gains or even cause decrements [21]. Training interventions that may be effective during the off-season often yield smaller adaptations in-season and can impose excessive additional load if not carefully managed [22]. When external resistance is added to repetitive overhead actions, as in volleyball spiking, the cumulative mechanical stress on the shoulder and elbow may further increase the risk of overuse syndromes if load and recovery are not properly controlled [6]. Hence, investigating wearable resistance in-season therefore provides insight into realistic coaching applications, but intervention design must be carefully balanced against existing competitive and training demands to avoid negative impacts on team performance.

Therefore, the aim of the study was to investigate the effect of an 8-week in-season training with wearable resistance attached to the forearm on spike velocity in female volleyball players. It was hypothesized that due to the gradual overload of forearm loading, while keeping the same spike technique, spike strength will increase, and thereby spike velocity over time will increase, as this was found in a similar study on throwing in handball players in competition season [15].

## 2. Materials and Methods

### 2.1. Experimental Design and Methodology

A pretest-post-test randomized controlled study was used to investigate the training effect of eight weeks of training with wearable resistance attached to the forearm in female volleyball players. A single senior women’s volleyball team was divided into two groups: one performed the intervention with wearable resistance attached to the forearms, while the other trained with wearable resistance to the calves. Using wearable resistance in both groups minimized Hawthorne effects, allowing a placement-specific comparison. A familiarization phase preceded both testing and the intervention to ensure safe and consistent execution. The study was conducted during the competitive in-season (October–December) with regular league matches on weekends. While players were aware of whether they wore forearm or calf wearable resistance, they were not informed of the specific study hypotheses. An overview of the study design and testing timeline is presented in Figure 1.

### 2.2. Participants

A total of 16 senior female volleyball players belonging to the same team (age 23.5 ± 3.2 years, body mass 66.8 ± 6.9 kg, and height 174.7 ± 5.8 cm) from the Polish second division were enrolled in the study. Participants were required to have at least three years of volleyball experience and to have actively competed within the past year. Exclusion criteria were current injuries that could impair testing or training performance, and training attendance below 80% (fewer than 13 of 16 sessions). The participants were fully informed about the procedures, the potential risks, and the benefits of the study, and all players provided written informed consent prior to participation. The study was conducted following the latest revision of the Declaration of Helsinki and was approved by the Norwegian Center for Research Data approved 12 November 2024 (project number: 775722).

Sample size determination: A priori power analysis in G*Power (v3.1) for a repeated-measures ANOVA (between–within interaction; α = 0.05; power = 0.80; *f* = 0.40; 2 groups × 2 measurements) indicated a required total sample of 16. All 16 eligible rostered players were enrolled; three withdrew, leaving *n* = 13 (forearm *n* = 7; calf *n* = 6).

### 2.3. Procedure

Testing was conducted one week before the start of the intervention and one week after its completion. First, a standardized warm-up was completed. After the standardized warm-up, each player performed eight maximal spikes during pre-testing to allow familiarization with the setup, and five spikes during post-testing once the procedure was already well learned. This difference was intentional to minimize fatigue during the in-season post-test while maintaining measurement reliability. Spikes were executed in a standardized manner: each player tossed to a teammate, received a set in front of the attack line, and performed a full approach jump and spike using a regulation volleyball. Trials were excluded if (1) the ball was mishit, producing a velocity clearly below the player’s normal range; (2) the attacking player signaled that the set or toss was of unacceptable quality; and (3) the radar failed to register a clean lock.

### 2.4. Measurements

The primary outcome was spike ball velocity, measured in km·h^−1^ using a Stalker Pro II+ professional sports radar (Applied Concepts, Richardson, TX, USA). The radar was positioned directly behind the attacker, aligned approximately at ball flight height, to minimize cosine error and ensure a clean signal. For analysis, the best three attempts from each session were averaged to emphasize near maximal performance while reducing the impact of occasional outliers or submaximal contacts. Testing took place in the same venue under identical conditions (same balls). All radar recordings were conducted by a single assistant operator, while the researcher supervised procedures to ensure consistency.

### 2.5. Training Intervention

After testing, the team was divided. Group allocation was determined using a computer-generated random sequence (www.randomizer.org). Wearable resistance was implemented using Lila sleeves (Lila™, Kuala Lumpur, Malaysia) designed for either the forearms or calves. Loads were attached bilaterally and progressed in absolute increments every two weeks. Weights were attached with Velcro on the arms, starting with 50 g per arm and increasing by 50 g every two weeks, reaching 200 g per arm by the end of the training intervention. This corresponded to 0.075%, 0.15%, 0.225% and 0.3% of body mass. The control group served as an active control group, wearing the wearable resistance sleeves (Lila™) on their calves (Figure 2) to ensure a comparable external load without directly affecting upper-limb mechanics (Table 1).

The training intervention was carried out from October to December, within the competitive season in which training exposure with wearable resistance consisted of ~30 min twice per week, embedded within regular team practice across the 8-week intervention. All subjects participated in the same volleyball training sessions led by the coach. Between each training session with wearable resistance was at least 48 h of rest, in addition to these, the team also completed two further weekly training sessions without wearable resistance, following their usual in-season schedule. Wearable resistance was applied during the initial part of practice when all players performed the same technical drills (e.g., approach, spike, and general team skills), ensuring standardized loading before the team split into position-specific training. No other modifications were made to the team’s in-season training plan, as the intervention was designed to test whether wearable resistance could be integrated without displacing normal practice content. All wearable resistance sessions were supervised by the researcher and team coach to ensure correct application, equipment safety, and adherence to the loading protocol.

### 2.6. Statistics

Data were checked for normality using the Shapiro–Wilk test, and equality of variances was assessed with Levene’s test. Descriptive statistics are presented as means ± standard deviations (SDs). Training effects of external weights on the forearm (experimental group) and calf (control group) were analyzed using a two-way repeated-measures analysis of variance (ANOVA) with factors time (pre-, post-test) and group (experimental vs. control) on maximal spike velocity. When significant differences were observed, post hoc tests were performed to locate pairwise differences. Effect sizes (*d*) were calculated using Cohen’s *d*, where *d* < 0.2 was considered small, 0.2–0.8 medium, and >0.8 large [23]. All analyses were conducted in JASP (version 0.18.1, Amsterdam, The Netherlands), and statistical significance was set at *p* < 0.05.

## 3. Results

Unfortunately, three participants dropped out during the intervention due to scheduling conflicts, leaving a final sample size of 13 participants (7 in the experimental group and 6 in the control group).

No significant main effect of group was found (*F*(1,11) = 3.12, *p* = 0.105), nor of time (*F*(1,11) = 0.06, *p* = 0.807). The interaction between group and time was also non-significant (*F*(1,11) = 0.01, *p* = 0.929), indicating that both groups changed similarly across the intervention (Figure 3). When analyzed separately, the forearm-loading group showed an average decrease in spike velocity of 2.1% (*p* = 0.14, *d* = 0.62; medium effect), while the calf-loading group decreased by 2.1% (*p* = 0.23, *d* = 0.31; small effect).

Two participants in the control group showed small increases in spike velocity (+0.1% and +4.0%), while two players in the experimental group improved by +10.3% and +2.4%. All other participants showed decreases ranging from −1.0% to −11.3% (Figure 4).

## 4. Discussion

The purpose of the study was to investigate the effect of eight weeks of forearm wearable resistance training on spike velocity in female volleyball players. The main findings were that spike velocity did not increase in either group, which was contrary to our hypothesis. In fact, neither the forearm-loaded experimental group nor the calf-loaded control group showed any performance improvement; both groups demonstrated small and similar declines in spike velocity over the 8-week period (on average, about 2.1% for both). These outcomes occurred with high individual variability, with no significant differences between the groups or across time. Given the small sample size, the statistical power to detect moderate effects was limited, and a Type II error cannot be ruled out.

Our rationale for this study was grounded in the specificity principle [18], with the expectation that overloading the striking arm would cause spike velocity to increase. However, the present findings suggest that the small fixed loads used (50–200 g per forearm) did not translate into measurable performance gains. From a biomechanical standpoint, attaching mass to the distal limb increases rotational inertia [8,20,24], which can slow angular velocities and potentially disrupt coordination unless sufficient adaptation occurs [24]. In this case, it appears that eight weeks was not enough time for positive neuromuscular adaptation to develop, as wearable resistance interventions shorter than ten weeks often yield limited changes in performance [9]. Overall, players maintained rather than improved their spike velocity, indicating that the added wearable resistance provided little net benefit under the present conditions. The discrepancy with the studies on overarm throwing can be explained by the training background. In these studies, college students, novices in handball [14,16] and lower-level experienced handball players [15] were used. It may be expected that novices and lower-level players have more potential to increase their performance faster, while experienced players, like those in our study, may already be near their performance ceiling or require a longer and more intense stimulus to elicit further gains.

Another likely contributor to the overall lack of improvement in spike velocity is the cumulative in-season fatigue and competing demands on the athletes. In the previous studies in handball, the players only trained twice per week [14,15,16], while in the present study, all players trained four times per week, thereby simultaneously exposed to the stresses of a competitive season, including frequent matches, travel, and limited recovery opportunities [21,22]. Such accumulated in-season load, including repeated accelerations, decelerations, and jump actions, has been shown to significantly contribute to fatigue in volleyball players [25]. These factors are well known to blunt or even reverse training adaptations during the season. In fact, it is normal for physical performance to stagnate or decrease slightly during a season as fatigue accumulates. Our control group’s modest drop in performance (~2% decline) is consistent with such an expected in-season effect [21,26]. The experimental group showed a similar small change (~2.1%), suggesting that the added arm loading did not meaningfully alter this seasonal trend. In practical terms, even a small loss of spike speed can be meaningful: spike velocity is a key determinant of attacking success in volleyball [1]. However, under the present conditions, the wearable resistance intervention did not appear to influence this outcome. Coaches should be cautious with extra-specific strength loading in-season, as any potential gains might be offset by fatigue and performance suppression, resulting in a neutral rather than beneficial outcome.

Although the group means did not change substantially, individual responses varied notably. One athlete in the forearm wearable resistance group stood out by increasing her spike velocity by about 6.5 km·h^−1^ (~10% improvement). Interestingly, this player was a receiver, a position demanding frequent serve receptions and defensive transitions, but not typically the primary attacker on the team. It is possible that her role-specific load and recovery status interacted positively with the intervention. For instance, as a receiver, she may have had fewer high-intensity spikes and thus less accumulated fatigue from matches, allowing her to benefit more from the wearable resistance training. Additionally, because she was not one of the strongest attackers initially, she may have had more room for improvement in spike velocity and responded well to the new stimulus. Meanwhile, several other players showed small decreases of 1–8 km·h^−1^, likely reflecting normal in-season variability rather than a clear negative training effect. This observation underscores the importance of presenting individual data when working with small samples, since interindividual variability in training response can substantially influence observed effects [27]. Each athlete’s adaptability can differ, and what impedes performance in one player might still enhance it in another under specific circumstances.

Of course, this study has some limitations. First, the sample was small due to the drop-out and use of a single team, which limits generalizability and statistical power. Although a repeated-measures design was used, the small sample increases the likelihood of Type II error and limits the strength of conclusions. However, by using a single team with an active control group drawn from the same team, we have avoided eventual training regime differences between teams as a confounding variable. Furthermore, the control group also trained with wearable weights (on the calves) to experience a similar novelty and training environment, which helps to minimize Hawthorne effects and isolate the influence of the forearm loading itself. Second, there was a lack of individualized load scaling, meaning all players used the same absolute weights (50–200 g) regardless of differences in body size or strength, so lighter or less trained players effectively received a greater relative stimulus than heavier or stronger players. This is supported by research showing that wearable resistance leads to individual differences in how whole-body coordination adapts under load [28]. Third, during the pre- and post-test a different number of spikes were performed, which one would think that this might induce fatigue. However, observation of the trials showed this was likely not an issue; several players achieved their very best velocity on the final attempt, suggesting that fatigue during testing did not confound the results. Finally, we measured only the outcome (ball velocity) and not the underlying kinematic or neuromuscular changes. It remains unclear whether the wearable resistance training induced any subtle technique adjustments or muscle activation changes that were not reflected in velocity.

Future studies should examine wearable resistance with larger cohorts, ideally across multiple teams, to improve statistical power and representativeness. Researchers should also test a broader range of loads relative to body mass and attachment sites (e.g., upper arm vs. lower arm). Incorporating mechanistic analyses, such as full kinematic assessments or electromyography (EMG) during spiking, would help to clarify when and how wearable resistance can best transfer to improve volleyball performance. For instance, analyzing the entire spike motion could reveal whether wearable resistance training alters technique in beneficial ways. By systematically exploring these factors, future research can determine strategies to optimize the use of wearable resistance for performance gains without incurring undue fatigue.

The present findings suggest that the use of small wearable loads on forearms in-season does not improve spike velocity and may even slightly reduce it under a heavy competitive schedule. Coaches should therefore be cautious when using this equipment, and they should prioritize recovery. If used in season, the loads are recommended to remain very light and applied in short, low-fatigue drills. It is possible that such training could be more beneficial during the off-season, when overall load is lower and players can better adapt to the added resistance, but controlled studies on off-season wearable resistance training in volleyball are currently lacking.

## 5. Conclusions

In this randomized, in-season trial with a single senior women’s squad, adding wearable resistance to normal practice did not increase spike velocity. Both placements, forearm and calf, showed small mean declines (~2.1%) with large individual variability. The between-group difference in change was small and not statistically significant. Taken at face value, this protocol does not support using wearable resistance during the competitive season to enhance spike velocity. Future work should examine whether off-season protocols, longer durations, or individualized load scaling might elicit more positive adaptations without adding fatigue.

## Figures and Tables

**Figure 1 jfmk-10-00458-f001:**
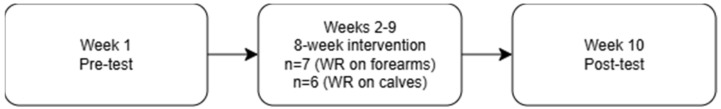
Overview of the experimental design and study timeline. Week 1: pre-test; Weeks 2–9: 8-week intervention with wearable resistance on forearms (*n* = 7) or calves (*n* = 6); Week 10: post-test.

**Figure 2 jfmk-10-00458-f002:**
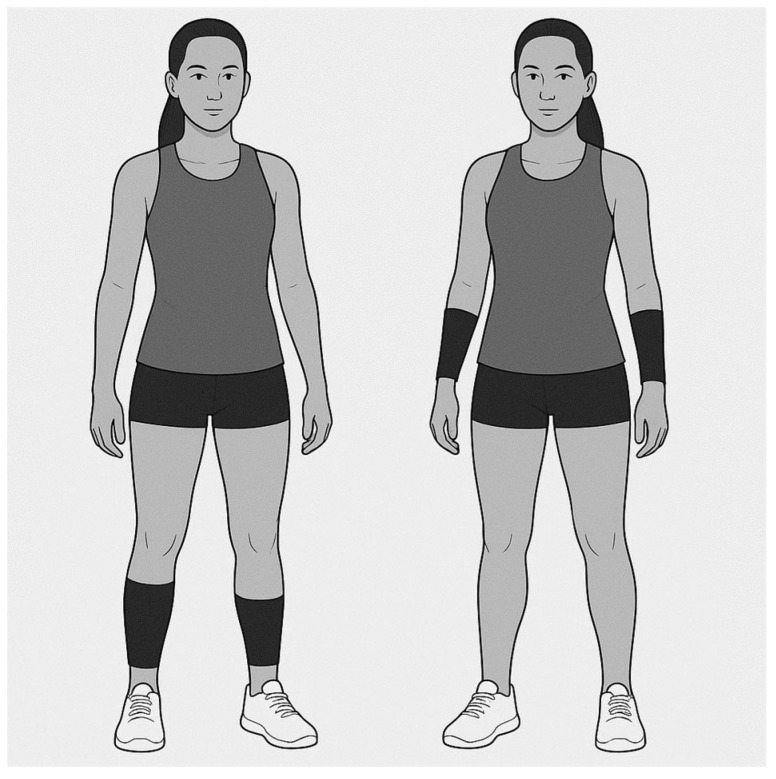
Illustration of the placement of wearable resistance. Typical WR load ranged from 50 to 200 g per limb for the forearms and 100 to 400 g per limb for the calves.

**Figure 3 jfmk-10-00458-f003:**
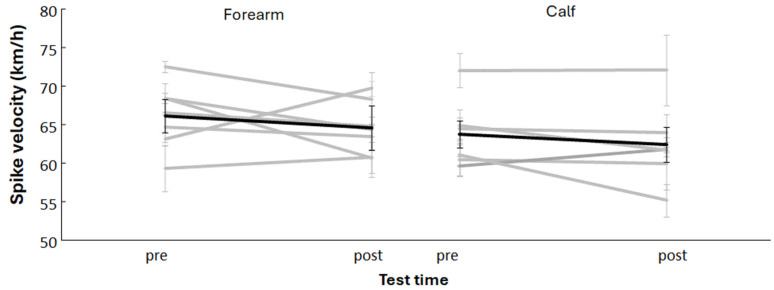
Maximal spike velocity for each subject and average (black lines) per experimental group (forearm) and control group (calf) on pre- and post-test.

**Figure 4 jfmk-10-00458-f004:**
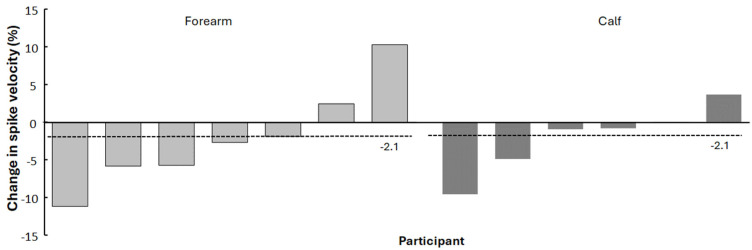
Percentage of change from pre- to post-test per participant in experimental (forearm) and control (calf) groups with average change per group (dotted lines).

**Table 1 jfmk-10-00458-t001:** The different weights placed on the forearm (experimental group) and calves (control group) per training session. Load progression corresponded to increments applied every two weeks.

Week	0	1		2		3		4		5		6		7		8		9
**Session**	Pre-test	1	2	3	4	5	6	7	8	9	10	11	12	13	14	15	16	Post-test
**Training weight (kg)**																		
**Forearm**		0.05				0.10				0.15				0.20				
**Calf**		0.10				0.20				0.30				0.40				

## Data Availability

The raw data supporting the conclusions of this article will be made available by the authors on request.

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
