# Peer review of "Effect of Eight Weeks of in Season Training with Wearable Resistance Attached to the Forearm on Spike Velocity in Female Volleyball Players"

_jfmk, 2025, doi:10.3390/jfmk10040458_

Round 1

Reviewer 1 Report

Comments and Suggestions for Authors

General comment 

The authors evaluated the effects of resistance training performed via wearable devices on specific technical skill execution (i.e., spike velocity) in volleyball players. The study covers an interesting topic and the procedures are described in detail. However, there are several points that should be clarified and discussed before it can be published.

Abstract

Line 10: please check the grammar fluency

Introduction

Line 65: Please consider to further provide the likelihood to get overuse upper-body syndromes due to repetitive overhead motions performed with an external resistance.

Methods

Line 82: please provide a graphical picture regarding the design setting

Line 92: please define how the actual sample size was determined, the inclusion and the exclusion criteria.

Results and Discussion 

Line 164: please add the standard deviation value within all figures

Line 252: please provide some practical applications, as well as future research perspectives derived from the Authors' results.

Author Response

Thank you very much for taking the time to review this manuscript. Please find the detailed responses below and the corresponding revisions/corrections highlighted/in track changes in the re-submitted files. We used COLOR for highlighting your responses in the text.

2. Point-by-point response to Comments and Suggestions for Authors

General comment

Comments 1:

The authors evaluated the effects of resistance training performed via wearable devices on specific technical skill execution (i.e., spike velocity) in volleyball players. The study covers an interesting topic and the procedures are described in detail. However, there are several points that should be clarified and discussed before it can be published.

Response: We appreciate the reviewer’s positive overall evaluation and the constructive feedback. All specific points raised have been carefully addressed in the revised manuscript. The requested clarifications and additions have been incorporated into the Abstract, Introduction, and Methods sections, and new text has been highlighted throughout the document.

Abstract

Comments 2:

Line 10: please check the grammar fluency

Response: The sentence in the Abstract was revised to improve grammatical fluency and clarity. The duplicated phrase “the effect of the effect of” was corrected, and the expression “an 8-week of in-season training” was simplified for readability.

Introduction

Comments 3:

Line 65: Please consider to further provide the likelihood to get overuse upper-body syndromes due to repetitive overhead motions performed with an external resistance.

Response: We thank the reviewer for this valuable suggestion. A clarifying sentence has been added in the Introduction to acknowledge that adding external resistance to repetitive overhead movements may increase mechanical stress on the shoulder and elbow, thereby elevating the risk of overuse syndromes.

Methods

Comments 4:

Line 82: please provide a graphical picture regarding the design setting

Response: We appreciate the reviewer’s suggestion. A schematic overview of the experimental design and study timeline has been added to the Methods section to visually present the testing sequence and intervention structure.

Comments 5:

Line 92: please define how the actual sample size was determined, the inclusion and the exclusion criteria.

Response: We thank the reviewer for this comment. The inclusion and exclusion criteria were already described in Section 2.2. In addition, a short paragraph has been added to explain how the sample size was determined. An a priori analysis in G*Power (v3.1; repeated-measures ANOVA, within–between interaction; α = 0.05; power = 0.80; expected f = 0.40) indicated a required total sample of 16. All 16 eligible team members were invited to participate, and three later withdrew, resulting in 13 participants.

Results and Discussion

Comments 6:

Line 164: please add the standard deviation value within all figures

Response: In figure 3 we have added the standard deviation. In figure we did not include this as it is the percentage of change for each subject, which is difficult to include a SD in the figure in our opinion.

Comments 7:

Line 252: please provide some practical applications, as well as future research perspectives derived from the Authors' results.

Response: We thank the reviewer for this suggestion. A paragraph has been added at the end of the Discussion section to outline the main applied implications for coaches and practitioners. This addition summarizes how wearable resistance can be integrated cautiously into in-season volleyball training and underlines the importance of managing fatigue and load to prevent negative performance effects.

Reviewer 2 Report

Comments and Suggestions for Authors

Major Comments

  1. While the sample is understandably small, it would be helpful to briefly mention the observed statistical power for the main outcome or discuss the risk of Type II error. This would contextualize the non-significant findings.
  2. Figures 2 and 3 are valuable and transparent. To enhance interpretation, consider including mean ± SD values for each group (pre and post) directly in the figure captions or a supplementary table.
  3. The Discussion is strong, but could briefly highlight whether small declines (–2 to –4%) in spike velocity are within the expected seasonal variability reported in volleyball performance monitoring literature. This would strengthen the practical interpretation.

Minor Comments

  1. Abstract – Minor repetition in the opening sentence (“effect of the effect of an 8-week…”). Please correct this duplication.
  2. Figure 1 caption – Add a short note on the typical absolute load range (e.g., “50–200 g per limb”).
  3. Methods (2.3 Procedure) – There is a small inconsistency in the number of spikes between pre-test (8 attempts) and post-test (5 attempts). Although later justified, you might want to clarify early in the section why this difference was intentional.
  4. Table 1 – Add a short explanation that the listed progression corresponds to increments every two weeks.
  5. Check spacing around hyphens and parentheses (e.g., “p <0 0.05” should read “p < 0.05”) (statistics)
  6. Ensure consistent formatting of statistical notation (e.g., “F = 3.91, p = 0.074” format used throughout).
  7. Conclusion – The conclusion is concise and relevant. You might consider adding one line suggesting that off-season interventions might yield different results, as a logical future direction.

Author Response

Thank you very much for taking the time to review this manuscript. Please find the detailed responses below and the corresponding revisions/corrections highlighted/in track changes in the re-submitted files. we used COLOR for highlighting your responses in the text.

2. Point-by-point response to Comments and Suggestions for Authors

Major Comments

  1. While the sample is understandably small, it would be helpful to briefly mention the observed statistical power for the main outcome or discuss the risk of Type II error. This would contextualize the non-significant findings.

Response: We thank the reviewer for this comment. A short statement has been added in the Discussion to acknowledge the limited statistical power and the potential for a Type II error due to the small sample size. This clarification helps contextualize the non-significant results.

  1. Figures 2 and 3 are valuable and transparent. To enhance interpretation, consider including mean ± SD values for each group (pre and post) directly in the figure captions or a supplementary table.

Response: In figure 3 we have added the standard deviation. In figure we did not include this as it is the percentage of change for each subject, which is difficult to include a SD in the figure in our opinion.

  1. The Discussion is strong, but could briefly highlight whether small declines (–2 to –4%) in spike velocity are within the expected seasonal variability reported in volleyball performance monitoring literature. This would strengthen the practical interpretation.

Response: We appreciate the reviewer’s observation. This point is already addressed in the Discussion, where we note that minor in-season declines in physical performance are expected and that our control group’s ~2% decrease is consistent with typical in-season effects [21,26]. We believe this adequately contextualizes the magnitude of change observed in the present study.

Minor Comments

  1. Abstract – Minor repetition in the opening sentence (“effect of the effect of an 8-week…”). Please correct this duplication.

Response: We thank the reviewer for noting this error. The duplication in the opening sentence of the Abstract has been corrected to improve grammatical fluency.

  1. Figure 1 caption – Add a short note on the typical absolute load range (e.g., “50–200 g per limb”).

Response: We thank the reviewer for this suggestion. The caption of the figure illustrating wearable resistance placement (now Figure 2) has been updated to specify the absolute load ranges: 50–200 g per limb for the forearms and 100–400 g per limb for the calves.

  1. Methods (2.3 Procedure) – There is a small inconsistency in the number of spikes between pre-test (8 attempts) and post-test (5 attempts). Although later justified, you might want to clarify early in the section why this difference was intentional.

Response: We thank the reviewer for pointing out this inconsistency. The sentence in Section 2.3 has been revised to clarify that eight trials were performed at pre-test for familiarization, while five trials were used at post-test to reduce fatigue once the athletes were fully accustomed to the procedure.

  1. Table 1 – Add a short explanation that the listed progression corresponds to increments every two weeks.

Response: We thank the reviewer for this helpful note. The caption of Table 1 has been updated to specify that the wearable resistance loads progressed in increments applied every two weeks.

  1. Check spacing around hyphens and parentheses (e.g., “p <0 0.05” should read “p < 0.05”) (statistics)

Response: We thank the reviewer for this careful observation. All statistical values have been reviewed, and spacing and notation have been standardized throughout the manuscript

  1. Ensure consistent formatting of statistical notation (e.g., “F = 3.91, p = 0.074” format used throughout).

Response: We thank the reviewer for this observation. Formatting, spacing, and notation now consistently follow the same structure.

  1. Conclusion – The conclusion is concise and relevant. You might consider adding one line suggesting that off-season interventions might yield different results, as a logical future direction.

Response: A sentence addressing the potential benefits of off-season wearable resistance interventions and the lack of existing research has been included at the end of the Discussion section.

Round 2

Reviewer 1 Report

Comments and Suggestions for Authors

The Authors addressed all suggestions properly. I don't have further comments.